# Remnant of the late Permian superplume that generated the Siberian Traps inferred from geomagnetic data

Shiwen Li [1], Yabin Li [1], Yanhui Zhang[2], Zikun Zhou[1], Junhao Guo[1] & Aihua Weng [1]✉

Mantle plumes have played a key role in tectonic events such as continental break-up and large magmatic events since at least the formation of Gondwana. However, as their signatures on Earth's surface, many of large igneous provinces have disappeared into the mantle during Earth's long-term evolution, meaning that plume remnants in the mantle are crucial in advancing mantle plume theory and accurately reconstructing Earth history. Here we present an electrical conductivity model for North Asia constructed from geomagnetic data. The model shows a large high-electrical-conductivity anomaly in the mantle transition zone beneath the Siberian Traps at the time of their eruption that we interpret to be a thermal anomaly with trace amounts of melt. This anomaly lies almost directly over an isolated low-seismic-wave-velocity anomaly known as the Perm anomaly. The spatial correlation of our anomaly with the Siberian Traps suggests that it represents a remnant of a superplume that was generated from the Perm anomaly. This plume was responsible for the late Permian Siberian large igneous province. The model strengthens the validity of the mantle plume hypothesis.

Large igneous provinces (LIPs) have formed at various sites around the world since the Palaeozoic[1], and the associated magmatic events have caused environmental catastrophes and mass extinction events[2,3]. It has been proposed that these LIPs resulted from mantle plumes that originated at the core–mantle boundary (CMB)[4]. However, only one-third of the catalogued hot spots on Earth is accompanied by a large columnar low-seismic-wave-velocity anomaly that reflects high temperatures indicative of a mantle plume[5]. Plume genesis remains controversial, even for the Columbia Plateau[6,7], which is the largest active LIP on Earth. With the exception of the Hawaiian oceanic island basalts (OIBs), the existence of a plume cannot be determined from the geochemical signatures of OIBs from hot spots[8], and this is a weakness of the mantle plume hypothesis. Another problem is the limited resolution of seismic imaging techniques[5]. However, seismic imaging has shown that some LIPs originated from superplumes[4,9] with broad heads (~1000 km in

diameter) near the mantle transition zone (MTZ)[6]. Furthermore, dynamic modelling has estimated that these superplumes are unusually persistent geological features[10,11]. Therefore, the identification of a plume head beneath a LIP could indicate the previous existence of a mantle plume. Only the roots of LIPs on continental plates are likely to be recognised, given the subduction of oceanic plates and the recycling of oceanic LIPs into the mantle[12]. Plume tails are ultimately assimilated into the ambient mantle as a result of mixing via mantle convection and energy dissipation after the cessation of the supply of hot material from the CMB[10]. These processes hinder our attempts to identify plume remnants.

The superplumes responsible for LIPs originate from plume generation zones (PGZs) surrounding two large low-shear-wave-velocity provinces (LLSVPs) at the CMB (Tuzo, beneath Africa; and Jason, beneath the South Pacific)[4]. These LLSVPs have been stable in their current locations throughout the Phanerozoic and possibly

[1]College of Geo-exploration Science and Technology, Jilin University, Changchun 130026, China. [2]School of Safety Engineering and Emergency Management, Shijiazhuang Tiedao University, Shijiazhuang 050043, China. ✉e-mail: wengah@jlu.edu.cn

longer[4,9]. Consequently, if a LIP whose site of formation has been restored can be spatially linked to a certain PGZ, then the trajectory of the plume feeding the LIP can be well constrained. The Siberian LIP, which formed during the late Permian and Early Triassic, is a suitable example for such an analysis[2,13]. This LIP was located over the Perm anomaly, a new-found small-scale low-shear-wave-velocity zone near the CMB[14]. The flood basalts of the Siberian Traps mainly cover the western Siberian rift system and Siberian Plateau[15,16], with the main volcanic pulse occurring over a period of <1 Myr at ~251 Ma[2,17]. Geochemical analysis has linked the production of the basalts in the rift system to the thermal effect of a plume after interacting with the MTZ, whereas the basalts covering the plateau were directly sourced from a CMB plume[16]. Numerical modelling suggests that the two plumes originated from a single CMB superplume[18,19].

A mantle plume is >300 K hotter[20] than the ambient mantle, is chemically distinct[8], and appears as a zone of low seismic wave velocity[21] and high electrical conductivity[22,23]. A plume that forms in the lower mantle has a narrow tail (<500 km diameter)[5]. However, when the plume impinges on the MTZ, the exothermic reaction due to the phase transition at the 660-km discontinuity[24] can result in the accumulation of a huge volume of hot plume material beneath the MTZ, potentially exceeding 1000 km in diameter[10]. While the plume tail may be below the resolution of seismic analyses or have diminished over time, the identification of a remnant mantle plume near the MTZ is crucial for inferring a past plume. The potentially large size of a remnant mantle plume near the MTZ should be detectable via the geomagnetic depth sounding (GDS) method[25,26]. This powerful method focuses on the slowly varying geomagnetic fields that are induced within the Earth, thus particularly suitable for identifying mantle conductivity anomalies at depths of 250–1600 km[25,26].

In this study, we utilize the GDS method to convert geomagnetic data in North Asia into a three-dimensional (3-D) electrical conductivity model. The model exposes a large high-electrical-conductivity anomaly in the MTZ beneath the Siberian Traps at the time of their eruption. By combining electrical conductivity with mineral physics modelling, the anomaly is interpreted as a thermal anomaly with trace amounts of melt. Our study provides electrical evidence for the remnant of a mantle plume that generated the late Permian Siberian large igneous province.

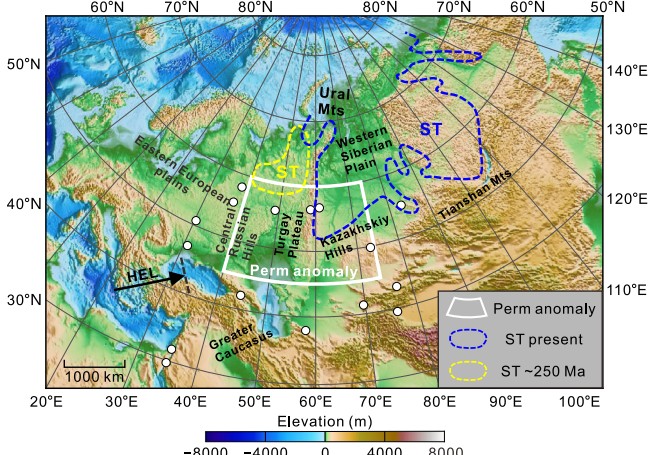

**Fig. 1 | Map showing the locations of geomagnetic stations and key features in the study area.** White dots indicate stations whose C-responses are used in this study. The range of Perm anomaly[14] is shown by zone enclosed by white lines. The current[2] (blue) and past[4] (yellow) locations of the Siberian Traps (ST) are indicated by dashed lines. The subduction direction of the Hellenic slab (HEL) is indicated by the black arrow, with the dashed black line representing the subduction front[39,66].

## Results

### Data and modelling

This study employs geomagnetic data from 16 observatories in North Asia (Fig. 1). We adopted the self-referenced method[25] to convert the observed signals to C-responses (Fig. S1). Most of the obtained responses are of good quality, with stable curves and high squared coherency (Fig. S2), which can be attributed in part to the continuous, ultralong records (>60 years at some observatories). The large C-response variations among the stations suggest significant heterogeneity in the electrical conductivity structure of the mantle beneath the study area. We converted the C-responses to electrical conductivity models of the mantle (250–1600 km depth) via a special 3-D inversion method[23] (see "Methods" for more details). The mantle above the CMB is divided into 12 spherical shells according to the one-dimensional (1-D) average global model[27]. Heterogeneous grid cells that span the Earth's surface are projected onto each shell, with the study area in the centre of the grid (Fig. S3). Although most of the observatories are far from the oceans, the ocean effect[25] is considered in our inversions. The model smoothing strategy embedded in the inversion yields the preferred model (Fig. S4) that minimises data overfitting (Fig. S5) and does not emphasise small-wavelength anomalies that are below the resolution of the data (Fig. S6).

### Electrical conductivity anomalies in the mantle

The most striking features in our preferred model are three high-electrical-conductivity anomalies in the MTZ and uppermost lower mantle beneath North Asia (Fig. 2 and S4). These anomalies are arranged like a string of beads along an almost west–east orientation. The strongest electrical conductivity anomaly is beneath the Perm area (hereafter termed the Perm electrical conductivity anomaly, or PEC). Geographically, the PEC is located beneath the southern margin of the Siberian Traps, stretching west from the Kazakhstan Hills through the Turgay Plateau to the Southern Plateau of the Ural Mountains. It extends ~1600 km east–west (50°E–70°E) and ~1000 km north–south (47°N–56°N). The PEC possesses average conductivities of 0.7 Sm$^{-1}$ in the MTZ and 2.0 Sm$^{-1}$ in the uppermost lower mantle; both values are approximately twice those of the average mantle[26]. The two secondary anomalies occur on the western and eastern flanks of the PEC. The eastern anomaly is located beneath the Kazakhstan Hills and its juncture with the Tianshan Mountains. It covers an area of 12° longitude × 6° latitude, with a conductivity value of up to ~0.8 Sm$^{-1}$ at its centre. The northwestern anomaly is beneath the Central Russia Hills on the Eastern European Plateau. The average conductivity of this anomaly is ~1.5 Sm$^{-1}$, and it covers an area of 20° longitude × 6° latitude. These two marginal anomalies are not discussed further in this study due to insufficient data.

The delineation of these anomalies, particularly the discovery of the PEC, can be attributed to the dense dataset and our special inversion technique[23]. Numerical tests of this approach indicate a resolution of ~9° longitude × ~9° latitude (~700 × ~1000 km) at MTZ depths (Fig. S6). This resolution is much finer than the PEC size, thereby highlighting the detectability of the PEC using current geomagnetic data. We are interested in the PEC due to its location beneath the Siberian Traps[4,13] and the lack of consensus on the deep origin of the traps[28,29]. We therefore conducted additional tests to verify the location and conductivity of the PEC (Figs. S7–S9), as the depth of the PEC coincides with the detection depth of the most reliable GDS data[26].

## Discussion

The high electrical conductivity in the MTZ might result from chemical composition, temperature, or the presence of partial melting and volatiles such as water[30,31]. The heterogeneous chemical composition of the MTZ has been attributed to the presence of subducted material[32]. A subducted slab that contains hydrous stishovite[33] and liebermannite[34] can have a high electrical conductivity[34,35] that is

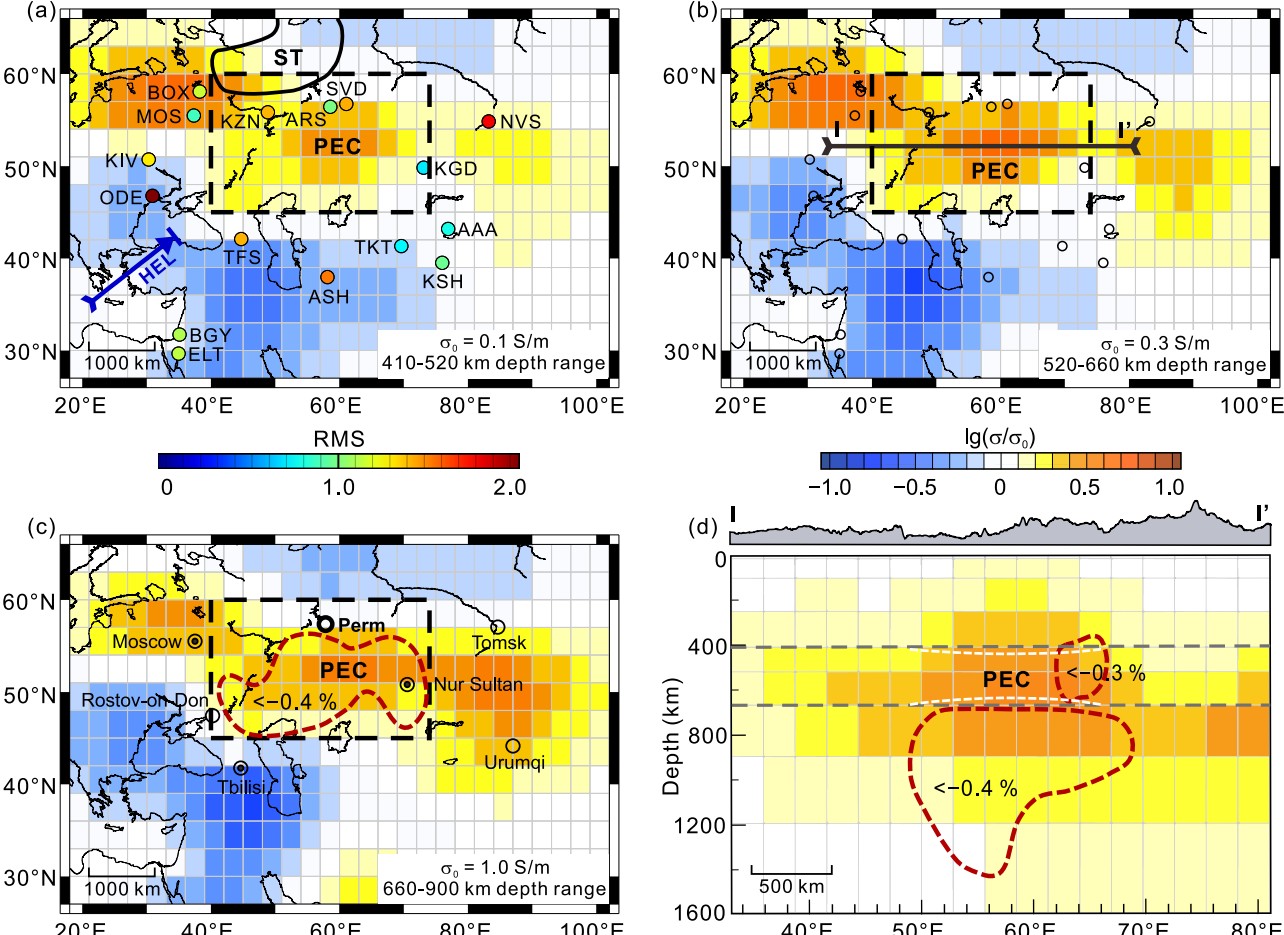

**Fig. 2 | Electrical conductivity images of the mantle beneath the study area.** Conductivity slices at depths of (**a**) 410–520 km, (**b**) 520–660 km and (**c**) 660–900 km. Coloured dots with three-letter codes in (**a**) represent the geomagnetic depth sounding (GDS) stations listed in Tab. S1, with the colours indicating the root mean square (RMS) of the data misfit at the end of the inversion. The areas enclosed by solid and dashed black lines denote the locations of the Siberian Traps (ST) when they formed[4] and the Perm anomaly near the core-mantle boundary[14],

respectively. The PEC denotes the Perm electrical conductivity anomaly. The blue line with an arrow indicates the subduction direction and front of the Hellenic (HEL) slab[39,66]. The black line labelled I–I' in (**b**) marks the location of the vertical section in (**d**). $\sigma_0$ is the global average conductivity. **d** Electrical structure beneath the I–I' profile along 51°N in (**b**). Dashed white lines, which are vertically exaggerated, show the depressed 410-km and elevated 660-km discontinuities[43]. Dashed red lines in (**c**, **d**) denote low-velocity bodies[40].

consistent with the observed high conductivity of the PEC. However, subducted slabs that possess high-velocity anomalies and were potentially associated with the Palaeo-Tethys Ocean and Mongol–Okhotsk Ocean[36] are not located near the PEC (Fig. 2a). Even if these palaeo-slabs did stagnate in the MTZ, they are up to 180 Myr in age[37] and therefore should have already avalanched into the lower mantle[38]. Indeed, a subducted component at the CMB directly below the PEC anomaly has been identified[2]. We also rule out this interpretation based on the modern Hellenic Plate (in terms of high-velocity)[39,40] (Fig. 1). And a low velocity band[40] about 10 degree south of the PEC accompanying the high velocities can strengthen this exclusion, as these low velocities are potentially due to the accumulation of the basaltic crust component detached from the plate[41], thus most likely representing the front of the plate.

The conductivity of minerals in the MTZ depends on their temperature and water content[30,31]. The PEC can be explained by the high water contents in wadsleyite (~1.5 wt.% at ~1800 K) and ringwoodite (~0.6 wt.% at ~1900 K) (Fig. S10), as the water filter mechanism[42] can trap large amounts of water. This explanation is in agreement with the observed velocity reduction of compressional seismic waves in the MTZ[40] (Fig. 2d), as water can reduce the seismic velocity[31]. However, the uplift of the 660-km discontinuity[43] (Fig. 2d) argues against the presence of water because water can depress the discontinuity, as

observed beneath the southwestern United States[44]. Alternatively, high temperatures can explain the observed velocity reduction and uplift of the discontinuity[21,45], and the conductivity anomaly observed in the present study. Similar observations of an uplifted discontinuity beneath the Hawaiian hotspot and hotspots in the South Pacific are explained by a superplume that originated from the Jason province[46]. Our conductivity modelling indicates that the temperature in the MTZ is ~250 K higher than the ambient mantle temperature (1900 K for 0.35 wt.% water in ringwoodite; 1800 K for 0.45 wt.% in wadsleyite)[30] (Fig. 3). Such a thermal anomaly is plausible because the PEC extends into the lower mantle (Fig. 2c), where the high conductivity requires an excess temperature of 350 K[22] (Fig. 3). This interpretation is supported by petrological data of the Siberian Traps[2,47].

In the lower mantle, the reduced density of the hot material[18] creates a positive buoyancy that can trigger a Rayleigh instability, which in turn leads to the upward movement of hot material. In the lower MTZ, the solidus of mantle minerals is ~2200 K for a water content of 0.35 wt.% or 2300 K for a water content of 0.2 wt.%[48]. This solidus is similar to the estimated temperature of the PEC, indicating the potential for partial melting. While direct measurement of the conductivity of melt in the MTZ remains challenging[49], we use the extremely high conductivity (~100 Sm⁻¹) of molten hydrated silicate ($Mg_2SiO_4$) (containing 14.3 mol.% water) under MTZ conditions[50] as a

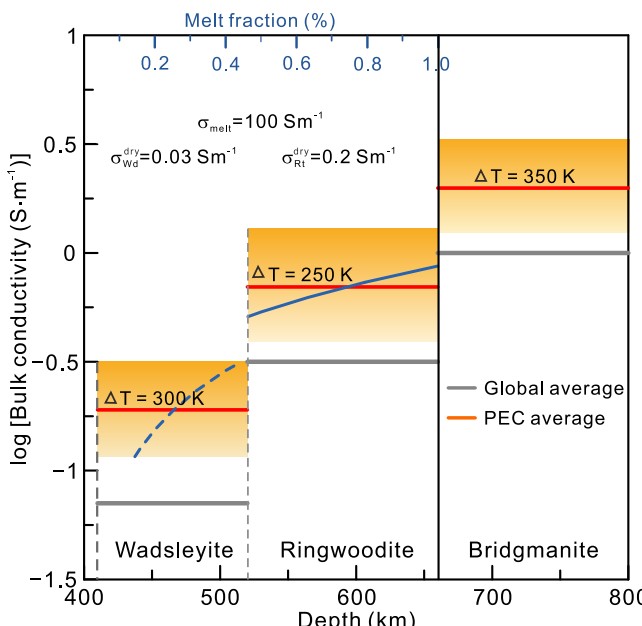

**Fig. 3 | Electrical conductivity of the PEC and interpretations of rock properties.** The conductivity of the Perm electrical conductivity anomaly (PEC) is indicated by the orange zones, with the thick red line denoting the average conductivity. The thick grey line denotes the global average conductivity model[27], which can be experimentally explained by wadsleyite and ringwoodite containing 0.45 and 0.35 wt.% water, respectively, in the mantle transition zone (MTZ) under normal geothermal conditions (ΔT = 0 K). The PEC in the upper and lower MTZ suggests excess temperatures of 300 and 250 K, respectively[30], while the PEC in the lower mantle can be interpreted to possess an excess temperature of 350 K[22]. Dashed and solid blue lines are the melting fractions of wadsleyite (Wd) and ringwoodite (Rt), respectively, based on the two-phase Hashin–Shtrikman upper–bound mixing model[64] and estimated from the MTZ bulk conductivity. The conductivity values of dry wadsleyite and ringwoodite[30], and hydrous silicate melts[50] are provided on the plot.

first-order approximation to estimate the melt fraction in the PEC, yielding <0.6 vol.% in the lower MTZ and ~0.2 vol.% in the upper MTZ (Fig. 3). This interpretation is consistent with the weak seismic velocity anomalies within and below the PEC (Fig. 2d), with high temperatures potentially inducing low velocities, although not to the extent that results from melt[51].

We conclude that the PEC represents the remnant of a plume that formed the Siberian Traps. Spatially, the PEC lies between a PGZ that is capable of generating the plume[4] and the Siberian Traps (~250 Ma) (Fig. 1). Therefore, the PEC can be considered a snapshot of the plume trajectory[16], and its anomalous excess temperature, which is potentially greater than ~300 K, provides a more direct signature for defining the mantle plume[5,8]. The low melting fractions indicates that melting is not straightforward in the MTZ, such that they may be residual material from the plume. The scale of the PEC indicate that the plume head below the 660-km discontinuity is ~1000 km across. This plume likely produced the Permo-Triassic Siberian Traps, which is the largest known LIP and whose basaltic rocks have enriched mantle II and prevalent recycled oceanic crustal components[2,11,13], and thus should be considered a superplume. This superplume likely generated secondary plumes[10,45,52]. The melting in these upwelling mantle plumes, aided by the migrating plume head, potentially triggered extensive partial melting in the upper mantle[11,51] and formed the moderate-altitude highlands in North Asia[18].

Taken together, the Perm anomaly, PEC and Siberian Traps constitute a textbook-like dynamic model of plume genesis at the CMB, especially in the case of interaction between the plume and the MTZ[10,45]. The plume would have impinged upon the 660-km

discontinuity after ascending from the Perm anomaly near the CMB, with the large Clapeyron slope (–2.0 MPa/K) at the interface providing sufficient resistance to prevent direct penetration of the plume through the MTZ[52,53], resulting in uplift of the 660-km discontinuity beneath the Siberian Traps[43]. This bridgmanite-to-ringwoodite phase boundary resistance, together with the weakened buoyancy[11] due to the dense recycled material[2], may have forced the plume to spread laterally beneath the interface[10], as imaged by the PEC. Plume melts are lesser dense in the lower mantle; therefore, more melts should have ultimately risen through the MTZ to the upper mantle and further produced the late Palaeozoic and early Mesozoic magmatism in North Asia[54]. The depression (by ~10 km) of the 410-km discontinuity[43] should be the sign that suggests that melts have passed through the discontinuity and entered the upper mantle (Fig. 2d).

Our electrical conductivity model provides no constraints on the timing of the plume[17,55]. However, the Clapeyron slope at MTZ interfaces and thermochemical structure within a plume that are critical in forming the PEC anomaly should be first-order parameters controlling the life of a plume. The life cycle of a superplume based on these parameters could be used for refs. [10,19]. Superplume ascent to the MTZ after formation at the CMB may take ~ (100–200) Myr likely with another up to ~100 Myr stalling in the upper mantle[11,56]. This is followed by a comparably short-lived flare-up that manifests as extensive magmatic activity, with the plume then entering its decay stage. Based on this time scale, the Siberian plume should now be in its waning phase, with the PEC being a remnant of this decaying superplume. The plume tail, owing to its thermal balance with the ambient mantle, may have been assimilated by the mantle or possess dimensions that are below the resolution of our data. The small Perm anomaly near the CMB[14] may represent a tail of the plume.

## Methods
### Data and materials
GDS is a passive electromagnetic method that can detect the electrical conductivity in and around the MTZ. The GDS C-response is defined as follows:

$$C(\omega) = -\frac{a_0 \tan\theta}{2} \frac{H_r(\omega)}{H_\theta(\omega)} \quad (1)$$

where $H_r$ and $H_\theta$ are the vertical and north-oriented components of the geomagnetic field, respectively; $\omega$ is the angular frequency; $\theta$ is the geomagnetic colatitude and $a_0$ is the Earth's radius[57]. The squared coherence (coh²) of $H_r$ and $H_\theta$ is commonly treated as a quality indicator of the C-response, which can detect sites with low noise-to-signal ratios[25]. The coh² is calculated as follows:

$$coh^2 = \frac{|\sum H_r W H_\theta^*|^2}{\sum H_r W H_r^* \sum H_\theta W H_\theta^*}, \quad (2)$$

where $W$ is an iterative robust weight matrix and the operator * is the complex conjugate. We modified the bounded influence remote reference processing (BIRRP) method[58] based on self-referenced technology[25] to estimate the C-responses from the observed geomagnetic fields. We carefully processed the records from 16 geomagnetic observatories in northern Asia (Tab. S1), and extracted their C-response curves for further consideration (Fig. S1). The C-responses have 16 periods that are logarithmically distributed over the range of 3.6–113.8 days. The response quality is good for most of the stations, as indicated by their large coherencies (Fig. S2). However, some stations have small coherencies (Fig. S2), indicating poor data quality or unstable responses. Therefore, the contribution from stations with low coherencies to the inversion is reduced by assigning very small weights to the responses, to ensure the reliability of the inversion results.

## Modelling and inversion

We used the limited-memory quasi-Newton method[23,59] to convert the observed C-responses to a 3-D electrical conductivity model. We parameterised the Earth using curved rectangular prisms into 12 spherical layers covering the core[27]. In contrast to global GDS inversions, where the horizontal grid is uniform[23,27], the study area in our inversions, which contains densely spaced observatories, was covered by a dense grid with a cell size of ~3° × ~3°. This cell size was progressively increased outside of the study area, reaching ~10° × ~10° to cover the entire globe (Fig. S3). The conductive Earth is parameterised to each prism, which yields a sufficiently high conductivity contrast and resolution[23]. A heterogeneous conductive surface layer with a thickness of 12.65 km was set to correct for the ocean effect[25,60] related to the large contrast in conductivity between ocean and land[27]. The background conductivity of each prism was obtained from Kelbert et al.[26] and iteratively inverted. We gradually reduced the regularisation parameter from an initial value of 100 to a relatively small value (10⁻⁴) to balance the data misfit and model roughness in our iterative inversion[26]. The variation in the root mean square (RMS) of the data misfit and other inversion parameters (regularisation factor, penalty function and model roughness) during the inversion iterations can assist in assessing the stability of our inversion (Fig. S5).

## Sensitivity tests

We performed numerical experiments to verify the prominent anomaly (Figs. 2, S4). We first used a plume model to evaluate the resolution of the inversion for our dataset. The inversion results revealed that our data could identify a plume-like anomaly with dimensions of ~10° × ~10° (Fig. S5). We then verified the robustness of the PEC (Figs. 2, S5) in two ways. First, we applied the model perturbation method. The presence of the anomalies in these models was corroborated by the obvious RMS changes at the stations near the PEC (Fig. S7). Second, we replaced the conductivity values in the anomalous region with those in the background model, and re-inverted our data with the preferred model as the initial model[23]. The reproducibility of the anomalies in the resultant models demonstrated that these features were required by the data (Figs. S8, S9).

## Electrical conductivity physics

The conductivity of mantle minerals is influenced by water content, temperature and the presence of melt[30,31]. The primary minerals in the MTZ, wadsleyite and ringwoodite, can contain relatively large amounts of water (up to 1–2 wt.%), while the most abundant mineral in the lower mantle, bridgmanite, apparently has a very low capacity of water storage[32,61]. Therefore, we attribute the enhanced conductivity to water and temperature in the MTZ, and temperature in the lower mantle.

The conductivities of hydrous wadsleyite and ringwoodite have been measured in the laboratory at various temperatures[30,62]. Figure S10 shows the conductivity adapted from Yoshino et al.[30]. According to the figure, a trade-off between water and temperature can explain the conductivity obtained by our model. The conductivity of bridgmanite that was measured by Sinmyo et al.[22] and Xu et al.[63] at lower mantle pressures can be used to reproduce our observed conductivity value.

The bulk conductivity ($\sigma$) of the MTZ containing partial melts can be calculated by the individual conductivities ($\sigma_1$ and $\sigma_2$, where $\sigma_2 > \sigma_1$) and volume fractions ($F_1$ and $F_2$) of the MTZ minerals and melts based on the Hashin–Shtrikman upper-bound (HS +) mixing model[64]:

$$\sigma = \sigma_2 \left[ 1 - \frac{3F_1(\sigma_2 - \sigma_1)}{3\sigma_2 - F_2(\sigma_2 - \sigma_1)} \right] \tag{3}$$

Almost all of the water in the MTZ minerals migrates into melts (phase 2) because the minerals/melt partitioning for water is rather small (on the order of 10⁻³)[59], resulting in a dry ambient MTZ (phase 1). Therefore, we treat the bulk conductivity in the MTZ as a function of hydrous melt, dry wadsleyite and ringwoodite, and their fractions. The conductivities of dry wadsleyite and ringwoodite are ~0.03 and ~0.2 Sm⁻¹, respectively[30]. We approximate the conductivities of molten wadsleyite and ringwoodite based on hydrous silicate melts under deep mantle conditions (~100 Sm⁻¹), given the lack of experimental data on their molten conductivities[50]. The bulk conductivity of the MTZ changes with the melting fraction, as shown in Fig. 3.

The base maps in Fig. 1, Fig. 2, Supplementary Fig. 3, Supplementary Fig. 4, Supplementary Fig. 7a, b, Supplementary Fig. 8a, b, and Supplementary Fig. 9a, b were all created using Generic Mapping Tools (GMT)[65].

## Data availability

Time series of the geomagnetic data used in this study can be accessed via the website www.wdc.bgs.ac.uk. C-responses data are provided in the Source Data file. Source data are provided with this paper.

## Code availability

Researchers interested in using code should contact S.L. (lisw1031@jlu.edu.cn).

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

## Acknowledgements

We thank Prof. Wenliang Xu at Jilin University and Prof. Jianfeng Yang from Institute of Geology and Geophysics, CAS for insightful discussions. This work was supported by the National Natural Science Foundation of China (42074080, 42130302 to A.W.; 42204076 to S.L. and 42104079 to Y.Z.) and the Program for Jilin University Science and Technology Innovative Research Team (2021TD-050 to A.W. and S.L.).

## Author contributions

A.W. and S.L. conceived the research. S.L., Y.L., and Y.Z. analysed the data. S.L., Z.Z. and J.G. produced figures. S.L. performed mineral physics modelling. A.W. and S.L. contributed to mineralogical discussion and final conclusion. A.W. and S.L. wrote the initial draft. All authors contributed to the preparation of the final manuscript.

## Competing interests

The authors declare no competing interests.
