## [Peer Review File · Nature Communications]

Remnant of the late Permian superplume that generated the Siberian Traps inferred from geomagnetic dataREVIEWER COMMENTS

Reviewer #1 (Remarks to the Author):

Review of Stephan Sobolev (consider as a signature). See attached file

Review

That is an interesting paper addressing a key problem of deep roots of LIPs. The key finding of this study is the cluster of the pronounced large-scale electric conductivity anomalies in the transition zone, with the strongest anomaly north of the Caspian Sea, that they call Perm Anomaly. Geographic location of this anomaly corresponding to the position of the Siberian LIP at 250Ma allows authors to associate this anomaly with the Siberian LIP. Because geographic location is so important for this interpretation, I think that the authors should discuss this point in more details. For instance, they mention that their anomaly corresponds to the P-waves velocity anomaly reported by Hosseini et al. (2020). However, looking in the details of the Hosseini's study, it is clear that although the cluster of the P-waves velocity anomalies is indeed similar in shape and orientation to the electric conductivity anomalies' cluster, it is located some 10 degrees further to the south, below the Black Sea and southern part of the Caspian Sea (see figure below).

I generally agree with the authors interpretation about thermal nature of their anomaly and that it may be related to the plumes and LIP at the surface. However, what about surface manifestation of the almost same intensive electric conductivity anomaly below the Eastern part of Russia; I think there was no evidence of strong volcanism in that area during the last 300 Myr.

Summing-up I think that this manuscript presents new interesting observations and suggest interesting, provocative interpretation. Therefore, I suggest publishing this paper in NC if the following conditions are met:

1. Technical part is OK, that I can not judge as being not an expert in electric conductivity.
2. Authors will discuss possible problems of their interpretation and particularly discrepancy with P-wave tomography in geographic location of the anomaly and absence of surface volcanism above other part of the anomaly.

Minor issues

Line165. Melts must not be less dense than solid in the transition zone. Moreover, if they could percolate through the matrix (low enough dihedral angle) they would concentrate either at the depth of neutral buoyancy or below the high viscosity barrier, and therefore will not contribute to the large-scale volume anomaly.

Line170. Brunet and Yuen (2000) study is nice but by no means recent. Ponding of the thermochemical plumes in transition zones were more recently modeled by Bullmer et al (2013 EPSL) and Dannberg and Sobolev (2015 Nature Comm.)

Reviewer #2 (Remarks to the Author):

That the Siberian Traps can be related to a plume sourced from the so-called Perm anomaly I am happy with (and has been stated a few times already in the literature), but there is a number of issues or arguments in his paper that concerns me to some extent. I must also confess that I am not an expert of electrical conductivity issues and will not comment on their methodology and analysis. The language also need improvements in some parts of the manuscript.

Specific comments/suggestions:

Abstract

Line 9: "However, many plumes have been recycled into the mantle during Earth's long-term evolution...":

What does this mean? Meaning plume-material like e.g. LIPs being recycled? Find to talk about e.g. ocean crust being recycled, but recycled plumes sounds a little strange.

Line 18: "We propose that this plume was responsible for the late Permian Siberian large igneous province":

Stated before in several papers as also said in the text. So "we propose" is not correct.

Line 19: "The model indicates the validity of the mantle plume hypothesis":

Replace the word "indicates" with "strengthen".

Introduction

Line 39: "However, seismic imaging has shown that some palaeo-LIPs originated from superplumes with broad heads (~1000 km in diameter) near the mantle transition zone (MTZ) and that are unusually persistent geological features, lasting for 400–600 Myr" :

There is absolute no evidence that these features can last 400-600 Myr (with reference to Brunet & Yuen 2000). Guess this is based on pure modelling and not observations?

Line 50: “The magmatic activity continued for ~30 Myr, with a peak at ~250 Ma^{2,13})”

Don't know where they get the idea that magmatic activity continued for about 30 Myr is. Does it mean Siberian Traps volcanism started at around 280 Ma and peaking at 251 Ma? That is probably not correct and literally all of the Siberian Traps were emplaced around 250 Ma. Use appropriate references like Augland et al. (2019). The main pulse of the Siberian Traps expanded in size and composition. Sci Rep 9, 18723. <https://doi.org/10.1038/s41598-019-54023-2>

Discussion

Line 154 “This plume, which likely produced the long-duration magmatism (>30 Ma) and the enriched mantle II and prevalent mantle components of the basalts in the Siberian Traps”:

See above my comment on long-duration magmatism.

Line 168 Our electrical conductivity model provides no constraints on the timing of the plume⁴⁸. However, recent dynamic modelling results have shown that an active superplume could persist for up to 600 Myr¹⁰”:

I seriously doubt these 600 Myr. Currently from observations we know that the most long-lived upwelling zone on Earth (currently the Tristan hotspot) has been active for say about 135 Myr.

Line 179: “..although dynamic modelling suggests that the anomaly is related to the subduction of slabs from the Mongol–Okhotsk and Palaeo-Tethys oceans”:

This is with reference to Flament et al. (2017) but in their more recent models they don't have any Perm anomaly any longer? So perhaps drop this argument. Also Torsvik & Domeier (2017) show that their 2017 paper is flawed.

Reviewer #3 (Remarks to the Author):

The EM profiles of the Earth's deep mantle are scarce; I appreciate the authors' efforts to utilize the recent advancements in the field of geophysics to investigate processes related to plume genesis.

The manuscript is well-written and logically organized. However, I have a few questions related to some of their interpretations; I believe that this might help clear some of the ambiguous discussions.

Considering the impact of these results on our understanding of the deep mantle plumes, I highly recommend the publication of this article in Nature Communications.

1. Plume component vs, deep-subducted slab : Authors argue that the high EC anomaly is related to the remnant plume component, not to a subducted slab passing through the MTZ. How confident are you in this argument? Can we directly co-relate the plume head to the erupted basalts by geochemical means? The fact that we have a subducted component at the CMB, directly below the alleged plume head, suggests this part could also be a part of that slab avalanched into the CMB. It might be useful to include some geochemical constraints that have been used in this conclusion.

2. Because of the high water storage capacities of major MTZ mineral phases, melting and melt migration are not so straightforward in this region. This fact is also corroborated by the seismic studies, which indicate low- seismic anomaly is only restricted to a small region. Especially, high EC is not correlating with the low-velocity anomaly. High T can induce low velocity but not to the extent, we have seen with melt.

The melting in the upwelling mantle, aided by the migrating plume-head, might seem to trigger melting at the upper mantle (please see Freitas et al. 2017 which discussed dehydration melting at the 410 km discontinuity, and Vinnik and Farra 2007).

3. Some of the missing references;

Freitas, D., Manthilake, G., Schiavi, F., Chantel, J., Bolfan-Casanova, N., Bouhifd, M. A., et al. (2017). Experimental evidence supporting a global melt layer at the base of the Earth's upper mantle. *Nat. Commun.* 8, 2186. doi:10.1038/s41467-017-02275-9.

Schmandt, B., Jacobsen, S. D., Becker, T. W., Liu, Z., and Ducker, K. G. (2014). Dehydration melting at the top of the lower mantle. *Science* (80-). 344, 1265–1268. doi:10.1126/science.1249850.

Vinnik, L., and Farra, V. (2007). Low S velocity atop the 410-km discontinuity and mantle plumes. *Earth Planet. Sci. Lett.* 262, 398–412. doi:10.1016/j.epsl.2007.07.051.

Freitas, D., and Manthilake, G. (2019). Electrical conductivity of hydrous silicate melts : Implications for the bottom-up hydration of Earth ' s upper mantle. *Earth Planet. Sci. Lett.* 523, 115712. doi:10.1016/j.epsl.2019.115712.

Minor comments: There seems to be a mismatch of references the lines 112-113

Geeth Manthilake

Dear Reviewers,
We would like to thank you for the constructive comments/suggestions, which have
significantly improved the manuscript. We have modified the manuscript according to these
comments. Please find our response to each reviewer comment/suggestions in blue. The line
numbers refer to the revised manuscript. All those modifications have all been highlighted in
red in this revised version.

Sincerely yours,

Aihua Weng and co-authors

-----

**Reviewer #1 (Remarks to the Author):**

**Review (consider as a signature).**

That is an interesting paper addressing a key problem of deep roots of LIPs. The key finding of
this study is the cluster of the pronounced large-scale electric conductivity anomalies in the
transition zone, with the strongest anomaly north of the Caspian Sea, that they call Perm
Anomaly. Geographic location of this anomaly corresponding to the position of the Siberian
LIP at 250Ma allows authors to associate this anomaly with the Siberian LIP. Because
geographic location is so important for this interpretation, I think that the authors should
discuss this point in more details. For instance, they mention that their anomaly corresponds to
the P-waves velocity anomaly reported by Hosseini et al. (2020). However, looking in the
details of the Hosseini's study, it is clear that although the cluster of the P-waves velocity
anomalies is indeed similar in shape and orientation to the electric conductivity anomalies'
cluster, it is located some 10 degrees further to the south, below the Black Sea and southern
part of the Caspian Sea (see figure below).

I generally agree with the authors interpretation about thermal nature of their anomaly and that
it may be related to the plumes and LIP at the surface. However, what about surface
manifestation of the almost same intensive electric conductivity anomaly below the Eastern
part of Russia; I think there was no evidence of strong volcanism in that area during the last
300 Myr.

Summing-up I think that this manuscript presents new interesting observations and suggest
interesting, provocative interpretation. Therefore, I suggest publishing this paper if the
following conditions are met:

[R1.1] 1. Technical part is OK, that I cannot judge as being not an expert in electric
conductivity.

[R1.2] 2. Authors will discuss possible problems of their interpretation and particularly
discrepancy with P-wave tomography in geographic location of the anomaly and absence of
surface volcanism above other part of the anomaly.

**Response:** Thank you for these comments and suggestions. They are very important for
improving the paper.

**Regarding** the reviewer's noted "discrepancy" between the geographic locations of the
P-wave tomography velocity anomaly and the PEC anomaly, we make the following points:

1. The velocity anomaly used in our paper has been extracted from the data provided by
Hosseini (downloaded from https://www.earth.ox.ac.uk/~smachine/cgi/index.php?page=tomo_depth). We plotted our electrical conductivity model and the Hosseini's velocity model by
using the GMT 6.0 tool. The slices at 520-600 km and 660-900 km are shown in **Fig. R1** for
comparison. We highlighted by red rectangles the regions of our anomalies. Overall, our
conductive anomalies correspond to the low velocity zones from Hosseini, thus ruling out the
possible problems associated with coordinate projection in figure preparing. In this regard, we
are particularly grateful to the reviewer for his consideration from the author's perspective.
Notably, owing to the difference in the methods to obtain the models, resolution of the two
resultant model is different. And our conductivity model is coarser than the velocity model
because only the data on 16 GDS stations are used.

**Fig. R1** Electrical (left panel) and velocity (right panel) anomalies at the MTZ depths (up row)
and uppermost lower mantle (bottom row).

2. There is a weak velocity anomaly ($\sim 0.4\%$) within the PEC anomaly area in the lower mantle,
as shown in Fig. R1, Fig. 2c and Fig. 2d along profile I-I'. The fundamental reason for this
difference in anomaly intensities is that the observed anomalies (electrical conductivity and
seismic velocity) are mainly caused by temperature variations, with the velocity effect being
less sensitive than the conductivity effect, particularly at small melting fractions (Freitas et al.,
2017). Reviewer #3 also made a similar comment regarding these two anomalies. This
comment has been incorporated into the revised text (Lines 148-150).

3. The P-wave velocity anomaly in the figure provided by the reviewer, for which the northern
boundary is marked with a dashed blue line, is an important and interesting finding that could
potentially constrain the subduction front of the Hellenic slab. This seismic anomaly is beyond
the scope of this paper, particularly since it lies outside of our study area and thus should not
co-relate to the PEC anomaly. However, using these anomalies, we could strengthen the
argument that the high conductivity of the PEC anomaly could not be caused by the Hellenic
slab. We therefore reworded the sentences in (Lines 121-123).

**Rewritten** (Lines 148-150): *This interpretation is consistent with the weak seismic velocity*
*anomalies within and below the PEC (Fig. 2d), with high temperatures potentially inducing low*
*velocities, although not to the extent that results from melt⁵⁰.*

**Rewritten** (Lines 121–123): *And a low velocity band⁴⁰ about 10 degree south of the PEC*
*accompanying the high velocities can strengthen this exclusion, as these low velocities are*
*potentially due to the accumulation of the basaltic crust component detached from the plate⁴¹,*
*thus most likely representing the front of the plate.*

**Regarding** the absence of surface volcanism above the anomaly in the eastern part of Russia:
We have identified the following potential reasons for the absence of volcanism over other
electrical conductivity anomalies. (i) The anomaly A in the reviewer's figure which corresponds
to the eastern part of Russia may have the same genesis as the PEC anomaly and may
produce materials of neutral buoyancy in the upper mantle, similar to the Siberian basalts,
which contain 10–20 vol.% eclogite (Sobolev et al., 2011). Magma storage in the upper mantle
may therefore only generate weak pre-magmatic surface deformation, as proposed by the
reviewer and Kirdyashkin & Kirdyashkin (2015). (ii) The age of the anomaly may differ from
that of the PEC anomaly (there are no age constraints on our electrical conductivity anomalies).
(iii) The electrical conductivity anomaly might be approximately the same age as the PEC
anomaly, but have different fates due to unknown factors. However, this electrical conductivity
anomaly, in combination with the lack of volcanism above it, is beyond the scope of our
knowledge, particularly since the formation mechanism of this anomaly is unclear. We also
note that these other anomalies are beyond the scope of this study, as our focus is on the PEC
anomaly; consequently, we do not discuss these anomalies in the manuscript.

**Minor issues**

[R1.3] Line165. Melts must not be less dense than solid in the transition zone. Moreover, if
they could percolate through the matrix (low enough dihedral angle) they would concentrate
either at the depth of neutral buoyancy or below the high viscosity barrier, and therefore will
not contribute to the large-scale volume anomaly.

**Response:** We agree with the reviewer in terms of the buoyancy of mantle melts as they
approach the MTZ. The melts associated with the Siberian plume may have stalled in the
upper mantle (Dannberg and Sobolev, 2015) due to the presence of eclogite (10–20 vol.%;
Sobolev et al., 2011). However, the plume should be lesser dense in the lower mantle (as
inferred by PEC, 350 K excess temperature in the MTZ), thereby allowing it to rise, with the
thermal effect being the main dynamic driver. However, the plume in its waning phase is weak,
and the presence of dense eclogite could weaken this dynamic force. The exothermic phase
transition at the 660-km discontinuity makes the interface work as a barrier. Collectively, all
these factors could produce a large anomaly near the base of the MTZ. These ideas have
been incorporated into the revised text (Lines 170–172).

**Rewritten** (Lines 170–172): *This bridgmanite-to-ringwoodite phase boundary resistance,*
*together with the weakened buoyancy¹¹ due to the dense recycled materia², may have forced*
*the plume to spread laterally beneath the interface¹⁰, as imaged by the PEC.*

[R1.4] Line170. Brunet and Yuen (2000) study is nice but by no means recent. Ponding of the
thermochemical plumes in transition zones were more recently modeled by Bullmer et al (2013
EPSL) and Dannberg and Sobolev (2015 Nature Comm.)

**Response:** Thank you for providing these important references. We have incorporated the
results from these more recent modelling studies into the text (Lines 35, 163, 170 and 182)
and referenced them accordingly.

-----

**Reviewer #2 (Remarks to the Author):**

That the Siberian Traps can be related to a plume sourced from the so-called Perm anomaly I
am happy with (and has been stated a few times already in the literature), but there is a
number of issues or arguments in his paper that concerns me to some extent. I must also
confess that I am not an expert of electrical conductivity issues and will not comment on their
methodology and analysis. The language also needs improvements in some parts of the
manuscript.

**Response:** We have provided point-by-point comments to address the reviewer's concerns,
and have incorporated the suggestions into the revised manuscript. The language has been
improved by a native English speaker.

Specific comments/suggestions:

Abstract

[R2.1] Line 9: "However, many plumes have been recycled into the mantle during Earth's

long-term evolution...”:
What does this mean? Meaning plume-material like e.g., LIPs being recycled? Find to talk
about e.g., ocean crust being recycled, but recycled plumes sounds a little strange.

**Response:** We agree with the reviewer’s suggestion. We mean that many large igneous
provinces, which are a surface expression of mantle plume material, have been recycled over
time (Lines 9-11). The text has been reworded for clarity.

**Rewritten** (Lines 9-11): *However, as their signatures on Earth’s surface, many of large*
*igneous provinces have subducted to the mantle during Earth’s long-term evolution, ...*

**[R2.2]** Line 18: “We propose that this plume was responsible for the late Permian Siberian
large igneous province”:

Stated before in several papers as also said in the text. So “we propose” is not correct.

**Response:** We agree with the reviewer’s suggestion and have deleted “We propose that” from
this sentence.

**Rewritten** (Lines 19–20): *This plume was responsible for the late Permian Siberian large*
*igneous province.*

**[R2.3]** Line 19: “The model indicates the validity of the mantle plume hypothesis”:

Replace the word “indicates” with “strengthens”.

**Response:** Revised.

**Rewritten** (Lines 20): *The model strengthens the validity of the mantle plume hypothesis.*

Introduction

**[R2.4]** Line 39: “However, seismic imaging has shown that some palaeo-LIPs originated from
superplumes with broad heads (~1000 km in diameter) near the mantle transition zone (MTZ)
and that are unusually persistent geological features, lasting for 400–600 Myr”:

There is absolute no evidence that these features can last 400-600 Myr (with reference to
Brunet & Yuen 2000). Guess this is based on pure modelling and not observations?

**Response:** We have estimated this time from the modelling results of Brunet & Yuen (2000),
as outlined in the figures therein. We have modified the text and deleted “lasting for 400 – 600
177 Myr” and added more references provided by reviewer #1 to strengthen the long duration of a
178 plume (Lines 34–35).

**Rewritten** (Lines 34-35): *Furthermore, dynamic modelling has estimated that these*
*superplumes are unusually persistent geological features^{10,11}.*

**[R2.5]** Line 50: “The magmatic activity continued for ~30 Myr, with a peak at ~250 Ma^{2,13}”

Don’t know where they get the idea that magmatic activity continued for about 30 Myr is. Does
it mean Siberian Traps volcanism started at around 280 Ma and peaking at 251 Ma? That is
probably not correct and literally all of the Siberian Traps were emplaced around 250 Ma. Use
appropriate references like Augland et al. (2019). The main pulse of the Siberian Traps
expanded in size and composition. Sci Rep 9,

18723. <https://doi.org/10.1038/s41598-019-54023-2>

**Response:** We agree with the reviewer's suggestion and have revised the text to constrain the
peak occurred at 251 Ma, and cited the provided reference (Lines 50–52).

**Rewritten** (Lines 50-52): *The flood basalts of the Siberian Traps mainly cover the western*
*Siberian rift system and Siberian Plateau^{15,16}, with the main volcanic pulse occurring over a*
*period of <1 Myr at ~251 Ma^{2,17}.*

Discussion

**[R2.6]** Line 154 “This plume, which likely produced the long-duration magmatism (>30 Ma) and
the enriched mantle II and prevalent mantle components of the basalts in the Siberian Traps”:
See above my comment on long-duration magmatism.

**Response:** We according to the above comments and the references provided by the
reviewer have modified this sentence (Lines 158–160).

**Rewritten** (Lines 158–160): *This plume likely produced the Permo-Triassic Siberian Traps,*
*which is the largest known LIP and whose basaltic rocks have enriched mantle II and prevalent*
*recycled oceanic crustal components^{2,11,13}, and thus should be considered a superplume.*

**[R2.7]** Line 168 Our electrical conductivity model provides no constraints on the timing of the
plume⁴⁸. However, recent dynamic modelling results have shown that an active superplume
could persist for up to 600 Myr¹⁰”:

I seriously doubt these 600 Myr. Currently from observations we know that the most long-lived
upwelling zone on Earth (currently the Tristan hotspot) has been active for say about 135 Myr.

**Response:** Thank you for the comment. Our meaning is that the whole life of a superplume
could persist for up to 600 Myr, including its active period which may be >135 Myr, as pointed
by the Reviewer. This life span from the results of Brunet and Yuan (2000) is used as a
reference because these results mainly take into accounts the effects on plume life of the
Clapeyron slope and plume thermal structure that are the main factors reflected by our model.
We also note that a number of factors can be employed to determine the lifespan of a plume.
For example, the neutral buoyancy of a plume, e.g., the Siberian plume, could create a plume
head that could persist for 100 Myr in the upper mantle (Dannberg and Sobolev, 2015), as
noted by reviewer #1. Further details of the modelling are beyond the scope of this paper. The
manuscript has been revised to reflect these comments (Lines 177–182).

**Rewritten** (Lines 177–182): *However, the Clapeyron slope at MTZ interfaces and*
*thermochemical structure within a plume that are critical in forming the PEC anomaly should*
*be first-order parameters controlling the life of a plume. The life cycle of a superplume based*
*on these parameters could be used for reference^{10,19}. Superplume ascent to the MTZ after*
*formation at the CMB may take ~ (100–200) Myr likely with another up to ~100 Myr stalling in*
*the upper mantle^{11,55}.*

**[R2.8]** Line 179: “...although dynamic modelling suggests that the anomaly is related to the
subduction of slabs from the Mongol–Okhotsk and Palaeo-Tethys oceans”:

This is with reference to Flament et al. (2017) but in the their more recent models they don't
have any Perm anomaly any longer? So perhaps drop this argument. Also Torsvik & Domeier

(2017) show that their 2017 paper is flawed.

**Response:** This argument has been removed from the manuscript.

-----

**Reviewer #3 (Remarks to the Author):**

The EM profiles of the Earth's deep mantle are scarce; I appreciate the authors' efforts to
utilize the recent advancements in the field of geophysics to investigate processes related to
plume genesis.

The manuscript is well-written and logically organized. However, I have a few questions
related to some of their interpretations; I believe that this might help clear some of the
ambiguous discussions.

Considering the impact of these results on our understanding of the deep mantle plumes, I
highly recommend the publication of this article in Nature Communications.

**[R3.1]** 1. Plume component vs, deep-subducted slab: Authors argue that the high EC anomaly
is related to the remnant plume component, not to a subducted slab passing through the MTZ.
How confident are you in this argument? Can we directly co-relate the plume head to the
erupted basalts by geochemical means? The fact that we have a subducted component at the
CMB, directly below the alleged plume head, suggests this part could also be a part of that
slab avalanched into the CMB. It might be useful to include some geochemical constraints that
have been used in this conclusion.

**Response:** Here we rule out the potential contribution from deep subducted slabs that may
have stagnated in the MTZ according to their positions identified by seismic imaging. However,
we agree with the reviewer that any subducted slabs that reached the CMB region may have
contributed to the formation of the PEC anomaly. We agree with the guidance provided by the
reviewer, and used the geochemical signature of recycled material (10–20 vol.%; Sobolev et
al., 2011) in the Siberian basalts as evidence that the PEC anomaly is derived from the lower
mantle. We have incorporated this comment into the text (Lines 118–119 and Lines 158–160).

**Rewritten** (Lines 118-119): *Indeed, a subducted component at the CMB directly below the*
*PEC anomaly has been identified².*

(Lines 158–160): *This plume likely produced the Permo-Triassic Siberian Traps, which is the*
*largest known LIP and whose basaltic rocks have enriched mantle II and prevalent recycled*
*oceanic crustal components^{2,11,13}, and thus should be considered a superplume.*

**[R3.2]** 2. Because of the high water storage capacities of major MTZ mineral phases, melting
and melt migration are not so straightforward in this region. This fact is also corroborated by
the seismic studies, which indicate low- seismic anomaly is only restricted to a small region.
Especially, high EC is not correlating with the low-velocity anomaly. High T can induce low
velocity but not to the extent, we have seen with melt.

The melting in the upwelling mantle, aided by the migrating plume-head, might seem to trigger
melting at the upper mantle (please see Freitas et al. 2017 which discussed dehydration
melting at the 410 km discontinuity, and Vinnik and Farra 2007).

**Response:** We agree that the electrical conductivity of the mantle is more sensitive to a
thermal anomaly than the seismic velocity of the mantle, as this thermal anomaly is strongly
dependent on the degree of melt (Freitas et al., 2017). We have primarily ascribed the PEC
anomaly to a thermal anomaly (Fig. 3), with only minor amounts of melt occurring in a small
region (Fig. 2d). This indicates that melting is not straightforward in the MTZ, such that the melt
may be residual material from the plume. More extensive melting in the upper mantle, which
was aided by the plume head, may have waned to the point of being undetectable and
insensitive to the data used in this study. We have incorporated these comments into our
interpretation of the PEC anomaly, and added the suggested references (Lines 148–150).

**Rewritten** (Lines 148-150): *This interpretation is consistent with the weak seismic velocity*
*anomalies within and below the PEC (Fig. 2d), with high temperatures potentially inducing low*
*velocities, although not to the extent that results from melt⁵⁰.*

**[R3.3]** 3. Some of the missing references;

Freitas, D., Manthilake, G., Schiavi, F., Chantel, J., Bolfan-Casanova, N., Bouhifd, M. A., et al.
(2017). Experimental evidence supporting a global melt layer at the base of the Earth's upper
mantle. Nat. Commun. 8, 2186. doi:10.1038/s41467-017-02275-9.

Schmandt, B., Jacobsen, S. D., Becker, T. W., Liu, Z., and Ducker, K. G. (2014). Dehydration
melting at the top of the lower mantle. Science (80). 344, 1265–1268.
doi:10.1126/science.1249850.

Vinnik, L., and Farra, V. (2007). Low S velocity atop the 410-km discontinuity and mantle
plumes. Earth Planet. Sci. Lett. 262, 398–412. doi:10.1016/j.epsl.2007.07.051.

Freitas, D., and Manthilake, G. (2019). Electrical conductivity of hydrous silicate melts:
Implications for the bottom-up hydration of Earth's upper mantle. Earth Planet. Sci. Lett. 523,
115712. doi:10.1016/j.epsl.2019.115712

**Response:** Thank you for providing these references. The missing references have been
added.

**[R3.4]** Minor comments: There seems to be a mismatch of references the lines 112-113

**Response:** This has been corrected.

REVIEWERS' COMMENTS

Reviewer #1 (Remarks to the Author):

Authors have addressed my comments generally well, so I would recommend the publication.

Corrections:

Abstract line 14, is "... beneath the Siberian Traps that.." should be "... beneath the Siberian Traps at the time of their eruption.."

Lines 48-50. Statement here is false. According Torsvik et al, 2014 (ref 4 in the ms.) reconstructions, the location of the main field of Siberian Traps at the time of their eruption (250Ma) corresponds to the of Perm anomaly, so in fact according to Torsvik et al, 2014 Perm anomaly is the plume generation zone (PGZ-abbreviation used by authors).

Line 149 "... interpretation is supported by isotopic data from Siberian basalts^{2,19}."

Neither Sobolev et al (2011) (ref. 2) nor Kirdyashkin and Kirdyashkin (2016) (ref 19) has used isotopic date to constrain excess temperature of Siberian Plume. Moreover I doubt that isotopic date could be used to do that.

Dear Reviewers,

We thank you again for the constructive comments, and we have modified the manuscript
according to these comments. Please find our response to the comment in blue. All those
modifications have all been highlighted in red in this revised version.

Sincerely yours,

Aihua Weng and co-authors

-----

**REVIEWERS' COMMENTS**

Reviewer #1 (Remarks to the Author):

Authors have addressed my comments generally well, so I would recommend the publication.

Corrections:

1. Abstract line 14, is "... beneath the Siberian Traps that.." should be "... beneath the Siberian
Traps at the time of their eruption.."

**Response:** We have corrected the manuscript.

**Rewritten (lines 13-15):** *The model shows a large high-electrical-conductivity anomaly in the
mantle transition zone beneath the Siberian Traps at the time of their eruption that we
interpret...*

2. Lines 48-50. Statement here is false. According Torsvik et al, 2014 (ref 4 in the ms.)
reconstructions, the location of the main field of Siberian Traps at the time of their eruption
(250Ma) corresponds to the of Perm anomaly, so in fact according to Torsvik et al, 2014 Perm
anomaly is the plume generation zone (PGZ-abbreviation used by authors).

**Response:** Thanks for the comment, and we have rewritten the sentence. Correspondingly,
the 'Tuzo Province' in lines 166 and 169 were replaced by 'Perm anomaly'.

**Rewritten (lines 48-49):** *This LIP was located over the Perm anomaly, a new-found
small-scale low-shear-wave-velocity zone near the CMB¹⁴.*

3. Line 139 "... interpretation is supported by isotopic data from Siberian basalts^{2,19}."
Neither Sobolev et al (2011) (ref. 2) nor Kirdyashkin and Kirdyashkin (2016) (ref 19) has used
isotopic date to constrain excess temperature of Siberian Plume. Moreover, I doubt that
isotopic date could be used to do that.

**Response:** Thanks for the comments, and we have modified the sentence and citation to
make the expression more accuracy.

**Rewritten (lines 138-139):** *This interpretation is supported by petrological data of the Siberian
Traps^{2,47}.*